# Assessment of Escape Safety of Cruise Ships Based on Dislocation Accumulation and Social Force Models

Jianing Li [1,2], Gaoshuai Wang [1], Yong Guo [1], Chao Liu [1], Yiming Huang [1,3] and Gang Chen [1,*]

1 Shanghai Waigaoqiao Shipbuilding Co., Ltd., Shanghai 200137, China
2 School of Naval Architecture, Ocean & Civil Engineering, Shanghai Jiao Tong University, Shanghai 200240, China
3 School of Transportation, Wuhan University of Technology, Wuhan 430062, China
* Correspondence: chengang@chinasws.com

**Abstract:** The escape safety of passengers is the primary design requirement of cruise ships. However, larger and more complex structural schemes make the existing escape safety assessment methods insufficient to accurately calculate the safety index of the passengers' escape. Therefore, this paper proposes a novel assessment method to infer the passengers' escape safety index of large cruise ships. Firstly, on the basis of quantifying personnel attributes and moving speed, a confluence algorithm based on the dislocation accumulation model is proposed to correct the flow model of passenger escape. Subsequently, a passenger escape flow method based on the social force model is established. The proposed method is applied to the escape safety assessment of a 130,000-ton cruise ship. The validation of the results is conducted by a comparative analysis between the proposed method and the traditional method and the results of simulation tools. The comparison confirmed that the proposed method has merits in computing accurate results. Overall, the proposed method supports the safety design of cruise ships.

**Keywords:** cruise ship; ship design; escape safety; safety assessment; escape time

## 1. Introduction

More extensive and complex cruise ships have emerged due to the explosion of the cruise ship market in the last decades, which, on the contrary, introduces more potential accidents to the sector [1–3]. The consequences of cruise ship accidents may be catastrophic, especially when those accidents happen at sea. For instance, the Costa Concordia cruise ship accident in 2012 resulted in 32 deaths. The reasons for the accident can be traced to the chaotic and unprofessional escape/evacuation process [2]. Therefore, an effective escape safety design scheme benefits for ensuring the safety of passengers [3], which is in the scope of the safety assessment [2,4–9], risk identification [3,10–12], and accident control [3,13–17] of complex systems.

Escape analysis refers to quantitatively evaluating the escape route, the performances of ladder arrangement, and other evacuation facilities during accidents like fires, which can determine the time required for evacuation and the escape safety level of cruise ships [18]. The International Maritime Organization (IMO) [19] issued an evacuation analysis framework for the escape safety assessment of cruise ships in the MSC/Cir.1001 guideline [20]. At present, the assessment of escape safety is mandatory for ship construction according to the IMO MSC/Cir.1533 guideline [21,22].

More recently, simulation-based and model-based methods have been applied to the escape analysis of cruise ships [23]. Specifically, the simulation-based method applies dedicated tools to obtain the time required for safe escape. Meanwhile, model-based methods analyze the evacuation time by constructing personnel movement time algorithms based on the personnel attributes and movement data.

The simulation-based methods simulate the movement of personnel and obtain evacuation time based on the attributes of passengers and the environmental characteristics of cruise ships [24]. For instance, Wang et al. [25] constructed an escape simulation model of a Ro-Ro cruise ship using CityFlow-M software, and the results are verified by the actual ship experimental data. Ginnis et al. [26] analyzed the escape time of a cruise ship by using the VELOS software, and based on the experimental data of three real ships, the results are verified. Kim et al. [27] carried out a simulation-based escape analysis on the basis of the IMEX software. The results confirmed that the software and its built-in algorithms have more advantages in simulating the movement speed of passengers than the other methods. Similarly, Guarin et al. [28] conducted a simulation and verification of the escape analysis of a cruise ship based on the EVI software. More scholars developed simulation tools, including Aeneas [29] and Marine EXODUS [30]. Marine EXODUS has been applied to the escape analysis of large cruise ships [31]. However, the safety assessment technology of the simulation-based method is restricted to the low flexibility of environment simulation models, which makes the input information of analysis limited to a certain environment. Therefore, one of the current trends in the field is to turn to model-based methods to extend the flexibility of the modeling and the mapping ability of the model to the actual escape process of cruise ships.

In terms of the safety escape models (model-based methods), Jin et al. [32] developed a new escape analysis model for cruise ships. The proposed model requires fewer input data than required by the MSC.1/Circ.1238 guide such as the width and length of stairs and corridors, and was proven to be easy to use. To improve the safety of cruise ships, Nasso et al. [33] applied the model-based method to calculate the escape time of a cruise ship carrying 3600 passengers and carried out a comparative analysis with the simulation-based method; the comparative results verified the correctness of the numerical results. Model-based escape analysis has also been used in the building industry. For example, Takeichi et al. [34] investigated the convergence process of the stairway in high-rise buildings and determined that the opening direction of the stairway and the state of the exit gate significantly impact the density of personnel in the stairway. Hokugo et al. [35] analyzed the convergence of passengers in stairways, the results showed that stairway width impacts the degree of crowding near stairs. Galea et al. [36] put forward a schedule for the arrangement of stairways and external platforms. The schedule has the potential to improve the convection efficiency of personnel. Zeng et al. [37] developed a safety evacuation model considering the escape of passengers to be three stages: free movement, extended zipper effect, and follow-up movement, which concluded the safe evacuation of a high-rise building. The results show that the concentration of the bottom personnel has a negative impact on the evacuation speed of upper personnel. Sano et al. [38] proposed a simplified mathematical model to calculate the evacuation time at the stairs and applied the modified model to the escape analysis of a 10-story building to determine the escape time under crowded conditions. By comparing it with the simulation-based method, the correctness and calculation efficiency of the proposed evacuation model is verified.

Compared with the simulation-based methods, the model-based escape analysis methods have the advantages of fast calculation and low cost. However, the following shortcomings need to be further improved: (i) The assumption of homogeneity of personnel (no difference among personnel) ignores the differences in the age composition, movement speed, and other parameters of passengers; hence, the calculated escape time has considerable uncertainty; (ii) The flow calculation method is difficult to simulate the multi-directional movement and convergence characteristics of passengers, which decreases the accuracy of results; (iii) The traditional methods are filed to simulate the flow characteristics under the congestion state during the escape process.

Aiming to overcome the aforementioned problems of model-based escape analysis methods, this paper proposes an escape safety assessment method based on dislocation accumulation and social force models to simulate the escape process of large-scale cruise ships. The novel contributions of this paper are as follows: (i) Determine the movement

speed of passengers considering the age and composition of passengers; (ii) Based on the dislocation accumulation algorithm, a two-way flow calculation method is constructed to map the flow of escaping passengers at the gathering point (stair area); (iii) According to the social force model, an escaping model is established to the assessment of the safety index like escape time of cruise ships under the crowded condition.

The rest of this paper is arranged as follows. Section 2 introduces the methodologies. Section 3 illustrates the results. Comparisons and validations are provided in Section 4. The conclusions are listed in Section 5.

## 2. Methodologies

### 2.1. Evacuation Time Model

According to the MSC.1/Circular.1533 issued by IMO, the escape time $T$ can be calculated by:

$$T = (\gamma + \delta) \times t_I + R \tag{1}$$

where $\gamma$ is the correction factor, $\delta$ is the counterflow correction factor equal to 0.3; $t_I$ is the maximum travel duration in ideal conditions, $R$ is the response duration, and it equals 10 min for the nighttime scenarios and 5 min for the day scenarios [22].

Generally, the escape sequence of passengers is from corridor to stairway and to the stairway exit where the muster station is located. The escape routes are considered to be a hydraulic network. To be specific, corridors and stairs are regarded as pipelines, and doors are treated as valves. Therefore, the time to pass an escape route until the assigned muster station is defined as:

$$t_I = t_F + t_{\text{deck}} + t_{\text{stair}} + t_{\text{assembly}} \tag{2}$$

where $t_F$ is the maximum duration of passengers within all stairways and corridors in an escape route; $t_{\text{deck}}$ is the travel duration(s) to move from the farthest point of the escape route of a deck to the stairway; $t_{\text{stair}}$ is the stairway travel duration(s) of the escape route to the assembly station; $t_{\text{assembly}}$ is the travel duration(s) to move from the end of the stairway to the entrance of the assigned muster station. Specifically, the calculation steps for the durations above are as follows:

(1) $t_{\text{deck}}$: Assuming that cabin passengers move into the corridor instantaneously at the same time, the personnel density ($D$) in the corridor area is decided by the area of the corridor and the number of people escaping [22]; The initial moving speed ($S$) of personnel is listed in Table 1. Then, the farthest escape distance and deck movement time (the ratio of the farthest escape distance to the initial movement speed of the personnel) of passengers in the cabin to the exit is calculated according to the corridor layout.

**Table 1.** Initial movement speed and flow of personnel.

| Facility | Initial Density $D$ (p/m$^2$) | Initial Density $F_s$ (p/(m $\times$ s)) | Initial Personnel Movement Speed $S$ (m/s) |
|---|---|---|---|
| | 0 | 0 | 1.2 |
| | 0.5 | 0.65 | 1.2 |
| Corridors | 1.9 | 1.3 | 0.67 |
| | 3.2 | 0.65 | 0.20 |
| | $\geq$3.5 | 0.32 | 0.10 |

(2) $t_F$: The flow time of passengers is the maximum flow time of each entrance and exit on the escape channel. The flow time of each entrance and exit is the ratio of the number of passengers to the flow. The passenger flow time in the corridor exit is based on the area of the corridor and the number of people, which is then used to calculate parameter $D$; According to the initial specific flow, $F_s$ of passengers, the ratio of the total number of people $N$ in the area to the specific flow, is determined. On the other hand, the flow time at the exit of the stairway can be computed according to the principle that has been shown in

Equation (3). It should be noted that the current analysis sets the maximum flow rate $Fs$ in different locations as listed in Table 2.

$$\sum F_c(in) = \sum F_c(out) \tag{3}$$

where $F_c(in)$ is the flow of route arriving at transition point; $F_c(out)$ is the route departing from transition point.

**Table 2.** Maximum flow in facilities.

| Facility | Maximum Flow $Fs$ (p/(m × s)) |
|---|---|
| Stairs (down) | 1.1 |
| Stairs (up) | 0.88 |
| Corridor | 1.3 |
| Gateway | 1.3 |

(3) $t_{\text{stair}}$: The flow in the stairway is determined by Equation (4). The moving speed ($S$) in the stairway refers to Table 3, in which the moving time in the stairway is the ratio of the length of the ramp to the moving speed, as:

$$t_{\text{stair}} = \frac{L_{ri}}{S} \tag{4}$$

where $L_{ri}$ is the length of the ramp of route $i$.

**Table 3.** Relationship between movement speed and flow.

| Facility | Specific Flow $Fs$ (p/(m × s)) | Movement Speed $S$ (m/s) |
|---|---|---|
| Stairs (down) | 0 | 1.0 |
| | 0.54 | 1.0 |
| | 1.1 | 0.55 |
| Stairs (up) | 0 | 0.8 |
| | 0.43 | 0.8 |
| | 0.88 | 0.44 |
| Corridor | 0 | 1.2 |
| | 0.65 | 1.2 |
| | 1.3 | 0.67 |

(4) $t_{\text{assembly}}$: Combined with the moving speed in the corridor under different flows in Table 3, the moving speed in the corridor to the gathering station can be obtained. The gathering time is the ratio of corridor length to moving speed, as:

$$t_{\text{assembly}} = \frac{L_{ci}}{S} \tag{5}$$

where $L_{ci}$ is the length of the corridor of route $i$.

However, the above model has the following shortcomings: (i) The initial flow and moving speed have a great impact on the calculation of the moving time. These inputs of the cruise ship's initial flow and initial movement speed are determined based on the civil building evacuation standard [8] and cannot perfectly map the evacuation scenarios of cruise ships; (ii) The calculation of the flow time $t_F$ in the stairway ignores the synchronicity of the movement of people on each deck, resulting in the calculated flow time being greater than the actual value; (iii) The impact of local congestion on the flow of people and the speed of movement is not clear. Accordingly, this paper proposed a new method to reflect the escape scenarios of cruise ships.

### 2.2. Proposed Computational Model

(1) Movement speed model.

According to MSC/Cir.1033, the composition of personnel on cruise ships is determined from the perspective of age and gender, as shown in Table 4. The movement speed of people in the corridor can be calculated according to the walking speed of age and gender function (Figure 1) and the proportion of cruise personnel, as shown in Equation (6). The movement speed of passengers in corridors is computed and listed in Table 5.

$$v_{c-w} = \sum_{i=1}^{n} w_i \times v_{c-i} \tag{6}$$

where $w_i$ is the proportion of passengers of different ages and genders, $v_{c-i}$ is the maximum or minimum movement speed of passengers in the corridor, $v_{c-w}$ is the weighted movement speed of passengers.

**Table 4.** Population composition on cruise ships [22].

| Population Groups—Passengers | Percentages (%) |
|---|---|
| Females < 30 years | 7 |
| Females 30–50 years old | 7 |
| Females > 50 years | 16 |
| Females > 50, mobility impaired (1) | 10 |
| Females > 50, mobility impaired (2) | 10 |
| Males < 30 years | 7 |
| Males 30–50 years old | 7 |
| Males > 50 years | 16 |
| Males > 50, mobility impaired (1) | 10 |
| Males > 50, mobility impaired (2) | 10 |

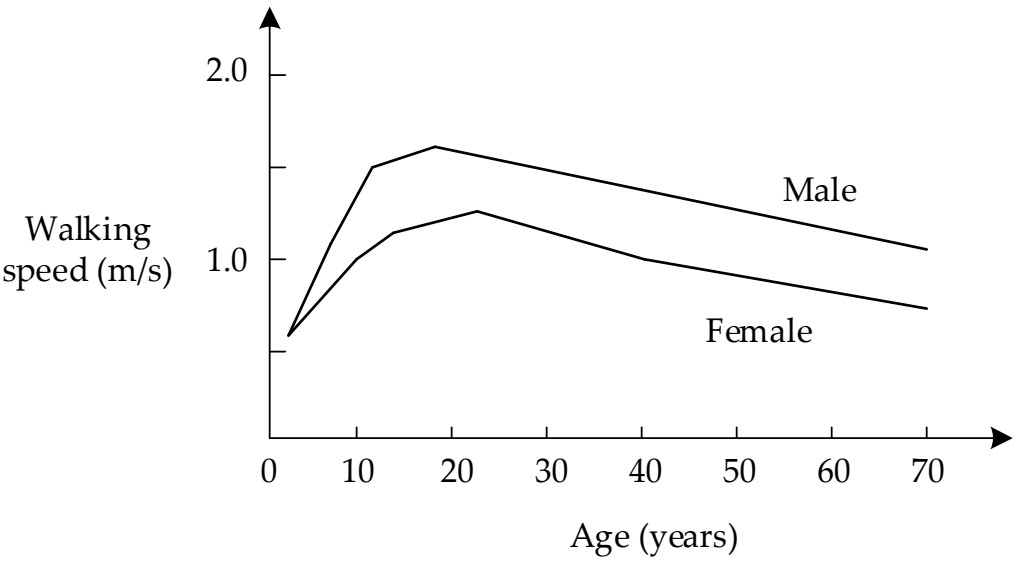

**Figure 1.** Movement speeds of passengers of different ages and genders.

This paper takes into account the movement speed of specific groups such as well-trained crew whose movement speed is significantly faster than passengers. According to the maximum and minimum values of the movement speed in Table 5, the movement speed in the corridor is corrected; see Table 6.

**Table 5.** Weighted movement speed of passengers in corridors [22].

| Passenger Groups | Weights | Min. Speed in Corridors (m/s) | Max. Speed in Corridors (m/s) |
|---|---|---|---|
| Females < 30 years | 0.07 | 0.93 | 1.55 |
| Females 30–50 years old | 0.07 | 0.71 | 1.19 |
| Females > 50 years | 0.16 | 0.56 | 0.94 |
| Females > 50, mobility impaired (1) | 0.1 | 0.43 | 0.71 |
| Females > 50, mobility impaired (2) | 0.1 | 0.37 | 0.61 |
| Males < 30 years | 0.07 | 1.11 | 1.85 |
| Males 30–50 years old | 0.07 | 0.97 | 1.62 |
| Males > 50 years | 0.16 | 0.84 | 1.4 |
| Males > 50, mobility impaired (1) | 0.1 | 0.64 | 1.06 |
| Males > 50, mobility impaired (1) | 0.1 | 0.55 | 0.91 |
| Weights | | 0.68 | 1.14 |

**Table 6.** The movement speed of passengers in corridor.

| Facility | Initial Density $D$ (p/m$^2$) | Conventional Used $S$ (m/s) | Corrected by the Proposed Method $S$ (m/s) |
|---|---|---|---|
| Corridors (Initial density) | 0 | 1.2 | 1.14 |
| | 0.5 | 1.2 | 1.14 |
| | 1,9 | 0.67 | 0.68 |
| | 3.2 | 0.20 | 0.2 |
| | ≥3.5 | 0.10 | 0.1 |
| Corridors (Flow) | 0 | 1.2 | 1.14 |
| | 0.65 | 1.2 | 1.14 |
| | 1.3 | 0.67 | 0.68 |

Based on the movement speed of passengers in the stairway given in Ref. [13] and the weight determined by the age of passengers as shown in Table 4, the maximum and minimum speeds of passengers when going up and down stairs in the stairway can be calculated using Equation (7), and the results are given in Tables 7 and 8.

$$v_{s-w} = \sum_{i=1}^{n} w_i \times v_{s-i} \tag{7}$$

where $v_{s-i}$ is the maximum or minimum movement speed of passengers of different ages and genders, $v_{s-w}$ is the movement speed of passengers in the stairway.

**Table 7.** Weighted a movement speed in stairway.

| Passenger Group | Weights | Min. Stairs Down Speed (m/s) | Max. Stairs Down Speed (m/s) | Min. Stairs Up Speed (m/s) | Max. Stairs Up Speed (m/s) |
|---|---|---|---|---|---|
| Females < 30 years | 0.07 | 0.56 | 0.94 | 0.47 | 0.79 |
| Females 30–50 years old | 0.07 | 0.49 | 0.81 | 0.44 | 0.74 |
| Females > 50 years | 0.16 | 0.45 | 0.75 | 0.37 | 0.61 |
| Females > 50, mobility impaired (1) | 0.1 | 0.34 | 0.56 | 0.28 | 0.46 |
| Females > 50, mobility impaired (2) | 0.1 | 0.29 | 0.49 | 0.23 | 0.39 |
| Males < 30 years | 0.07 | 0.76 | 1.26 | 0.5 | 0.84 |
| Males 30–50 years old | 0.07 | 0.64 | 1.07 | 0.47 | 0.79 |
| Males > 50 years | 0.16 | 0.5 | 0.84 | 0.38 | 0.64 |
| Males > 50, mobility impaired (1) | 0.1 | 0.38 | 0.64 | 0.29 | 0.49 |
| Males > 50, mobility impaired (2) | 0.1 | 0.33 | 0.55 | 0.25 | 0.41 |
| Weights | | 0.46 | 0.76 | 0.36 | 0.60 |

**Table 8.** Weighted average movement speed in stairway.

| Type of Facility | Specific Flow *Fs* (p/(m × s)) | Conventional Used *S* (m/s) | Revised by the Proposed Method *S* (m/s) |
|---|---|---|---|
| Stairs (down) | 0 | 1.0 | 0.76 |
|  | 0.54 | 1.0 | 0.76 |
|  | 1.1 | 0.55 | 0.46 |
| Stairs (up) | 0 | 0.8 | 0.6 |
|  | 0.43 | 0.8 | 0.6 |
|  | 0.88 | 0.44 | 0.36 |

(2) Flow model.

Takeichi [34] studied the personnel concentration process in the stairway of high-rise buildings. Through the personnel concentration experiment, the influence of the personnel density in the stairway, the opening direction of the stairway, and the state of the exit door on the stairway concentration are analyzed. Sano [38] proposed a mathematical model for calculating the evacuation time of the stairway, considering the impact of the personnel concentration, to evaluate the impact of the concentration of people entering the stairway and people in the stairway on the pedestrian flow and evacuation time. The current cruise flow calculation model based on the flow accumulation method uses the same stairs to escape from each deck. The circulation time is the cumulative sum of the flow of passengers, and the flow rate ratio is computed by:

$$t_{\mathrm{F-stair}} = \frac{\sum\limits_{i=1}^{n} N_i}{F_s} \tag{8}$$

where $t_{\mathrm{F-stair}}$ is the duration that passengers move within the stairway, $N_i$ is the number of passengers who enter the stairway, $n$ is the number of decks where people enter the escape ladder, $F_s$ is the personnel flow.

Passengers on all decks escape at the same time in case of fire on a cruise ship. When the upper deck passengers move down along the stairway, the middle deck passengers have two movement directions: horizontal movement along the corridor and downward movement through the stairway, i.e., only some middle deck passengers meet with passengers from upper decks at the entrance of the stairway, and the rest of the passengers transfer to the muster station. The current flow-accumulation-based flow calculation model is difficult to map the above actual scenarios. Therefore, this paper proposes a new flow calculation algorithm based on dislocation accumulation, as:

$$t'_{\mathrm{F-stair}} = \frac{\sum\limits_{i=a}^{b} N_i - \left( \sum\limits_{i=a}^{b-1} \left[ F_{C_i} \times t_i \right] \right)}{F_s} \tag{9}$$

where $t'_{\mathrm{F-stair}}$ is the duration that a person flows through a stairway using a misplaced add-on algorithm, and a, b decks are the lowest and the uppermost deck of the escape using the stairway. The b-1 deck is the lower deck of the b deck. $N_i$ is the number of passengers who enter the stairway, and $F_{C_i}$ is the flow of passengers who move in advance of the convection of the deck except for the upper deck. $t_i$ is the flow duration of the upper deck before it reaches the deck. For example, in the case of b-1 decks, the number of passengers moving in advance on b-1 decks is the result of multiplying the time required to move the human flow from "b" deck stairways to b-1 deck stairways and the flow on b-1 deck.

In the proposed dislocation accumulation algorithm, it is assumed that the movement time difference of each deck in the corridor is small before entering the stairway, and the passengers on each deck arrive at the stair entrance of a specific floor at the same time. Therefore, when passengers on the upper deck move into the stairway, the passengers on the lower deck before the confluence should be considered.

(3)    Traffic calculation model under congestion.

Traditional escape analysis methods based on local rigidity theory consider that the space between passengers is incompressible, and it is difficult to map the flow characteristics under congestion [21,22]. Therefore, Helbing et al. [39,40] proposed a social force model based on the concept of social force to establish a micro pedestrian simulation model. Moreover, Helbing [41] summarized the social force model and proposed the influence of individual behaviors. Li [42] proposed a method for the quantitative analysis of pedestrian congestion during evacuation based on the Hughes model and social force model theory, considering the spread and change of pressure between people during evacuation. Therefore, based on the personnel flexibility theory in the social force model, this paper establishes a traffic calculation algorithm under congestion. It holds that a passenger is an elastic body, and the squeezing pressure of individuals will increase the sliding friction between individuals in a crowded state. Based on the above, the proposed traffic calculation model under local congestion is expressed as follows:

$$
F_s = \begin{cases} \dfrac{\sum\limits_{i=1}^{n} F_{C_i}}{w} & 0 \le \dfrac{\sum\limits_{i=1}^{n} F_{C_i}}{w} \le 1.3 \\[4ex] \dfrac{1.3^2 \times w}{\sum\limits_{i=1}^{n} F_{C_i}} & 1.3 \le \dfrac{\sum\limits_{i=1}^{n} F_{C_i}}{w} \end{cases} \tag{10}
$$

wher, $F_{C_i}$ is the flow entrancing into the congested area, w is the exit width of the traffic in the congestion area, and the coefficient (1.3 p/(m $\times$ s)) is the maximum of the corridor flow.

It should be noted that the personnel flexibility theory is applicable to platforms (such as corridors), and the passengers in stairway space are obviously staggered up and down. Therefore, the above model ignores the extrusion characteristics of the human body in the ladder movement environment.

## 3. Results

### 3.1. The Ship and Data Sources

The ship for the escape analysis in this paper is a 130,000 gross tonnage cruise ship with the capability of carrying 5958 passengers. The external view of the cruise is shown in Figure 2. The Main Vertical Zone (MVZ, No. 1), the MVZ (No. 4), and the MVZ (No. 5) are three important MVZs, and the escape route arrangement of the three MVZs are independent. The maximum horizontal escape distance, number of cross deck floors, vertical escape distance, and length of assembly route of the three main vertical zones are illustrated in Table 9.

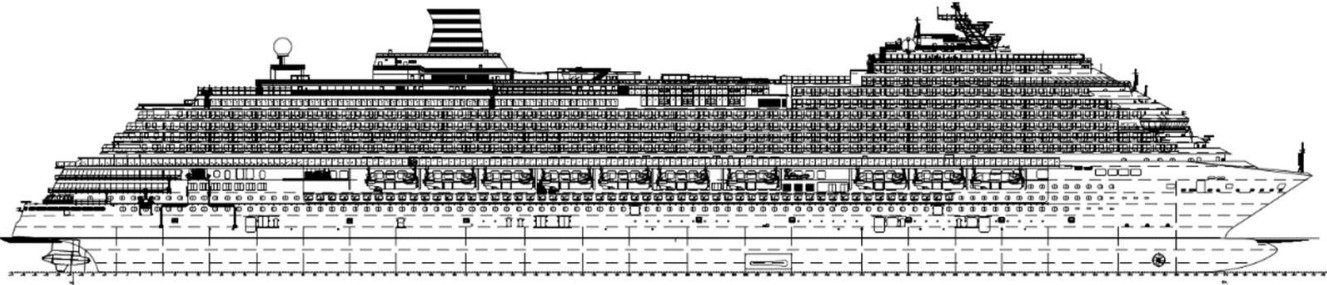

**Figure 2.** External view of the cruise ship.

The basic scenarios for the escape analysis include daytime and nighttime scenarios. Compared with daytime scenarios, people are distributed in the cabins, and the escape distance is longer. Therefore, nighttime scenarios are selected for this analysis. According to the general layout of the cruise ship and the escape route design, the passenger allocation under the night scenario is tabulated in Table 10.

**Table 9.** Escape parameters of each main vertical zone.

| Main Vertical Zone Parameters | MVZ 1 | MVZ 4 | MVZ 5 |
|---|---|---|---|
| Maximum horizontal escape distance (m) | 30.3 | 46.1 | 36.1 |
| Number of cross deck layers (layer) | 6 | 6 | 6 |
| Vertical escape distance (m) | 31.8 | 33.4 | 33.7 |

**Table 10.** The allocation of personnel in the night scene in MVZs.

| Deck | Night Case in MVZ 1 | | Night Case in MVZ 4 | | Night Case in MVZ 5 | |
|---|---|---|---|---|---|---|
| | Passengers | Crew | Passengers | Crew | Passengers | Crew |
| 1 | 128 | 18 | 124 | 2 | 109 | 2 |
| 2 | 225 | 2 | 139 | 2 | 108 | 2 |
| 3 | – | – | – | – | – | – |
| 4 | – | 10 | – | 10 | – | 10 |
| 5 | 87 | 4 | – | 18 | – | 10 |
| 6 | 118 | 2 | 106 | 1 | 63 | 2 |
| 7 | 94 | 2 | 104 | 2 | 96 | 2 |
| 8 | 94 | 2 | 104 | 2 | 92 | 2 |
| 9 | 79 | 2 | 100 | 2 | 95 | 2 |

### 3.2. Results of the Proposed Method

In combination with the passenger distribution and corridor and stairway layout, the initial characteristics of personnel movement are shown in Table 11, in which $C$ is the *ID* of corridor, $D$ represents the *ID* of door, and $S$ denotes the *ID* of stairway in the ship. The calculated escaping time of MVZ 1, MVZ 4, and MVZ 5 are listed in Table 12.

**Table 11.** Personnel flow and movement speed.

| Deck | Facility ID | Initial Persons | $D$ (p/m²) | $F_S$ [p/(m × s)] | $S$ (m/s) | $F_C$ (p/s) |
|---|---|---|---|---|---|---|
| 1 | C111 | | | | | 1.3 |
| | D111 | 66 | 0.9 | 0.8 | 1.0 | 1.2 |
| | S111 | | | | | 0.8 |
| | C112 | 64 | 0.9 | 0.8 | 1.0 | 1.3 |
| | D112 | | | | | 1.2 |
| | C113 | | | | | 0.8 |
| | D113 | 16 | 0.7 | 0.7 | 1.1 | 0.8 |
| | S112 | | | | | 0.8 |
| 2 | C211 | | | | | 1.5 |
| | D211 | 111 | 1.1 | 0.9 | 0.9 | 1.5 |
| | S211 | | | | | 1.3 |
| | C212 | 108 | 1.1 | 0.9 | 0.9 | 2.0 |
| | C213 | | | | | 0.5 |
| | D212 | 8 | 0.3 | 0.4 | 1.1 | 1.7 |
| | S212 | | | | | 1.3 |

**Table 12.** Travel durations by conventional method.

| Deck | MVZ 1 | | | | | MVZ 4 | | | | | MVZ 5 | | | | |
|---|---|---|---|---|---|---|---|---|---|---|---|---|---|---|---|
| | $t_{stair}$ (s) | $t_{deck}$ (s) | $t_F$ (s) | $t_I$ (s) | $T$ (s) | $t_{stair}$ (s) | $t_{deck}$ (s) | $t_F$ (s) | $t_I$ (s) | $T$ (s) | $t_{stair}$ (s) | $t_{deck}$ (s) | $t_F$ (s) | $t_I$ (s) | $T$ (s) |
| 1 | 23.2 | 44.4 | 301.7 | 369.3 | 849.4 | 21.4 | 59.0 | 100.8 | 181.3 | 416.9 | 23.1 | 40.4 | 45.5 | 109.1 | 250.9 |
| 2 | 12.1 | 103.9 | 301.7 | 417.7 | 960.7 | 12.5 | 67.5 | 100.8 | 180.9 | 416.1 | 14.0 | 39.8 | 45.5 | 99.3 | 228.4 |
| 4 | 18.6 | 0.0 | 301.7 | 320.3 | 736.6 | 12.8 | 0.0 | 100.8 | 113.7 | 261.5 | 12.8 | 0.0 | 49.0 | 61.8 | 142.2 |
| 5 | 28.6 | 43.7 | 301.7 | 374.0 | 860.3 | 25.7 | 0.0 | 100.8 | 126.5 | 291.0 | 26.1 | 0.0 | 49.0 | 75.1 | 172.8 |
| 6 | 40.4 | 42.4 | 301.7 | 384.6 | 884.5 | 39.2 | 65.9 | 128.6 | 233.7 | 537.6 | 39.7 | 38.1 | 49.0 | 126.7 | 291.3 |
| 7 | 49.6 | 40.1 | 301.7 | 391.4 | 900.2 | 51.0 | 67.2 | 128.6 | 246.7 | 567.5 | 51.4 | 40.6 | 49.0 | 141.0 | 324.3 |
| 8 | 58.8 | 71.3 | 301.7 | 431.8 | 993.2 | 61.8 | 66.9 | 128.6 | 257.3 | 591.9 | 62.3 | 39.9 | 49.0 | 151.1 | 347.6 |
| 9 | 68.1 | 38.9 | 301.7 | 408.7 | 940.0 | 70.2 | 84.2 | 128.6 | 283.0 | 650.9 | 69.0 | 40.4 | 49.0 | 158.3 | 364.2 |

## 4. Discussion

The results of the proposed method are compared with the conventional one based on the MSC/Cir.1533 guideline. To analyze the differences between the two methods, the total escape time of each route should be calculated and compared. The escape routes contained in the three main vertical zones are shown in Table 13.

**Table 13.** Duration of MVZ 1/4/5 based on conventional method.

| MVZ | No. | Route Code | No. | Route Code | No. | Route Code | No. | Route Code | No. | Route Code |
|---|---|---|---|---|---|---|---|---|---|---|
| | 1 | MVZ1-E1 | 6 | MVZ1-E6 | 11 | MVZ1-E11 | 16 | MVZ1-E16 | 21 | MVZ1-E21 |
| | 2 | MVZ1-E2 | 7 | MVZ1-E7 | 12 | MVZ1-E12 | 17 | MVZ1-E17 | 22 | MVZ1-E22 |
| MVZ 1 | 3 | MVZ1-E3 | 8 | MVZ1-E8 | 13 | MVZ1-E13 | 18 | MVZ1-E18 | 23 | MVZ1-E23 |
| | 4 | MVZ1-E4 | 9 | MVZ1-E9 | 14 | MVZ1-E14 | 19 | MVZ1-E19 | | |
| | 5 | MVZ1-E5 | 10 | MVZ1-E10 | 15 | MVZ1-E15 | 20 | MVZ1-E20 | | |
| | 1 | MVZ4-E1 | 4 | MVZ4-E4 | 7 | MVZ4-E7 | 10 | MVZ4-E10 | 13 | MVZ4-E13 |
| MVZ 4 | 2 | MVZ4-E2 | 5 | MVZ4-E5 | 8 | MVZ4-E8 | 11 | MVZ4-E11 | 14 | MVZ4-E14 |
| | 3 | MVZ4-E3 | 6 | MVZ4-E6 | 9 | MVZ4-E9 | 12 | MVZ4-E12 | | |
| | 1 | MVZ5-E1 | 4 | MVZ5-E4 | 7 | MVZ5-E7 | 10 | MVZ5-E10 | 13 | MVZ5-E13 |
| MVZ 5 | 2 | MVZ5-E2 | 5 | MVZ5-E5 | 8 | MVZ5-E8 | 11 | MVZ5-E11 | 14 | MVZ5-E14 |
| | 3 | MVZ5-E3 | 6 | MVZ5-E6 | 9 | MVZ5-E9 | 12 | MVZ5-E12 | 15 | MVZ5-E15 |

### 4.1. Comparison of Stairway Movement Time

It can be seen from Figures 3–5 that the movement time based on the proposed method is larger than that of the conventional one. The reason is that the proposed method considers the impact of age distributions of passengers so that the speed of movement becomes longer in stairways, which is more in line with reality and computing a safer and more convincing result.

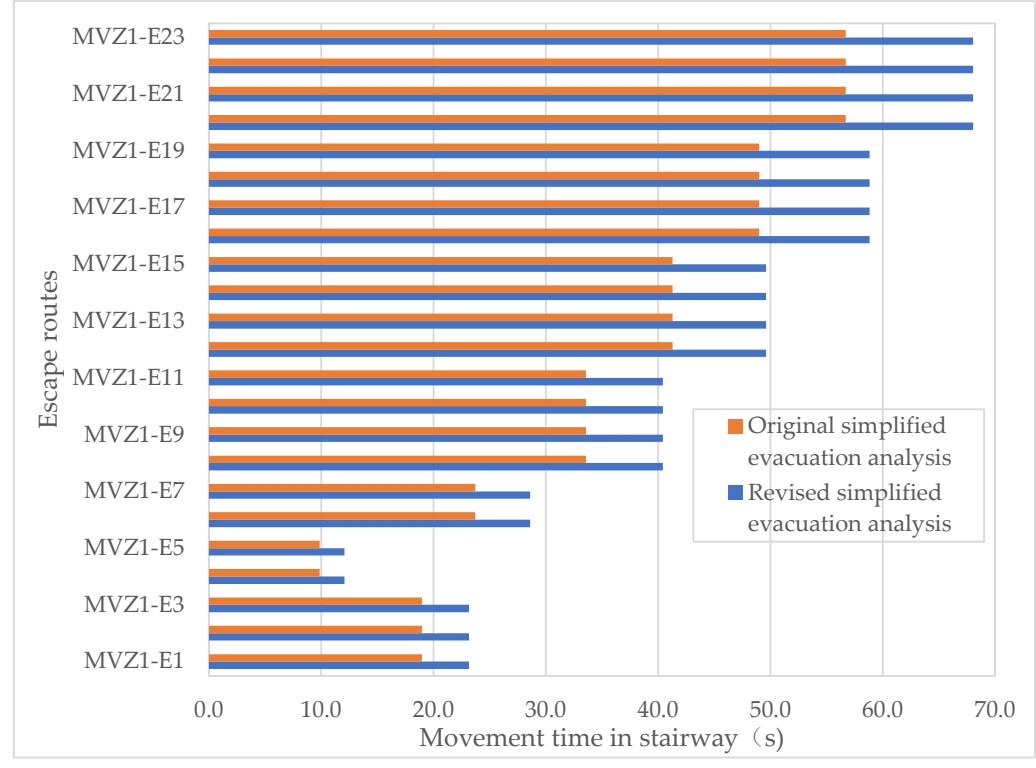

**Figure 3.** Comparison of movement time in the stairway in MVZ 1/Original simplified evacuation analysis method: MSC/Cir.1533; Revised simplified evacuation analysis method: The proposed method in this paper.

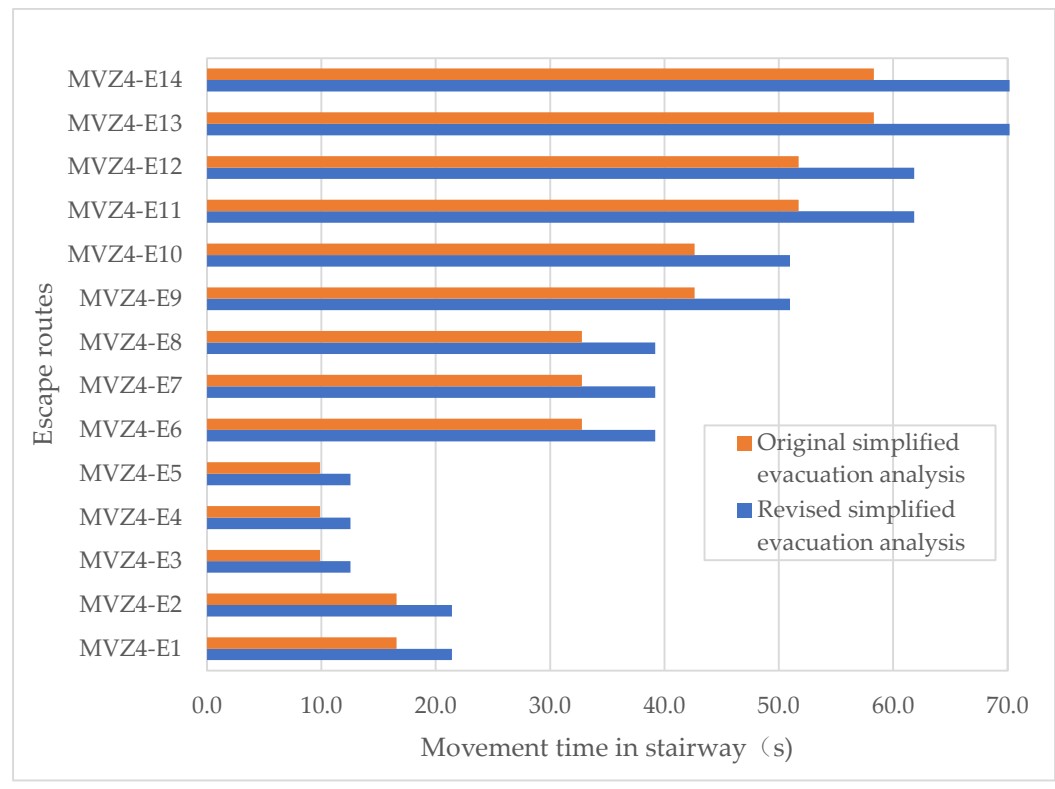

**Figure 4.** Comparison of movement time in the stairway in MVZ 4/Original simplified evacuation analysis method: MSC/Cir.1533; Revised simplified evacuation analysis method: The proposed method in this paper.

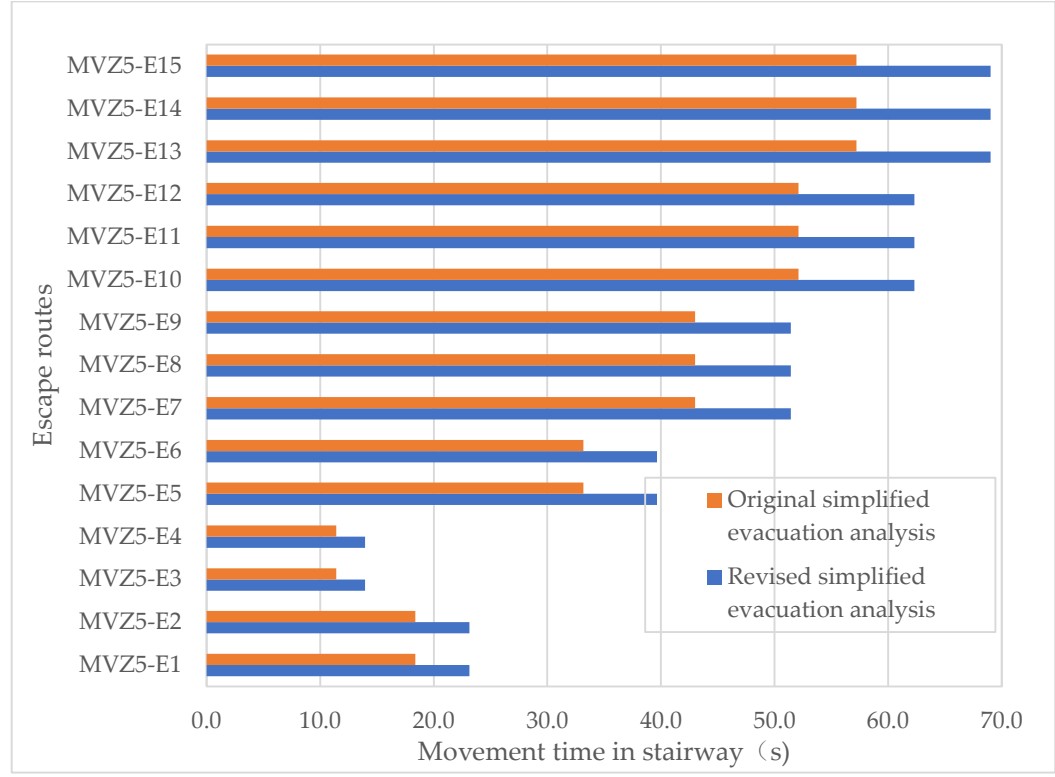

**Figure 5.** Comparison of movement time in the stairway in MVZ 5/Original simplified evacuation analysis method: MSC/Cir.1533; Revised simplified evacuation analysis method: The proposed method in this paper.

### 4.2. Comparison of Flow Time

According to the proposed flow algorithm in the stairway, the passengers on board will receive the public address system alarm simultaneously in case of fire. Therefore, part of the passenger on the deck escaped earlier and did not participate in the escaping flow of the various decks. As a result, the flow time of passengers in the stairway is smaller than that computed by the conventional method. At the same time, the proposed local congestion method based on the social force model defined a lower flow in the state of extreme congestion, which decreases the flow speed of passengers when moving to congested doors. The calculations based on both methods are shown in Figures 6–8. As shown in Figures 6–8, the flow time of the No. 1, No. 4, and No. 5 main vertical zones are calculated by using the original model-based method and the model-based method proposed in this paper. The flow time of the method proposed in this paper is 25%, 20%, and 33% lower than that of the original method. In other words, the flow speed in the method proposed in this paper is higher than that of the original model. This is mainly because the dislocation accumulation model established in this paper reduces the flow of people in the convergence of the stairway, which increase the flow speed in the stairway.

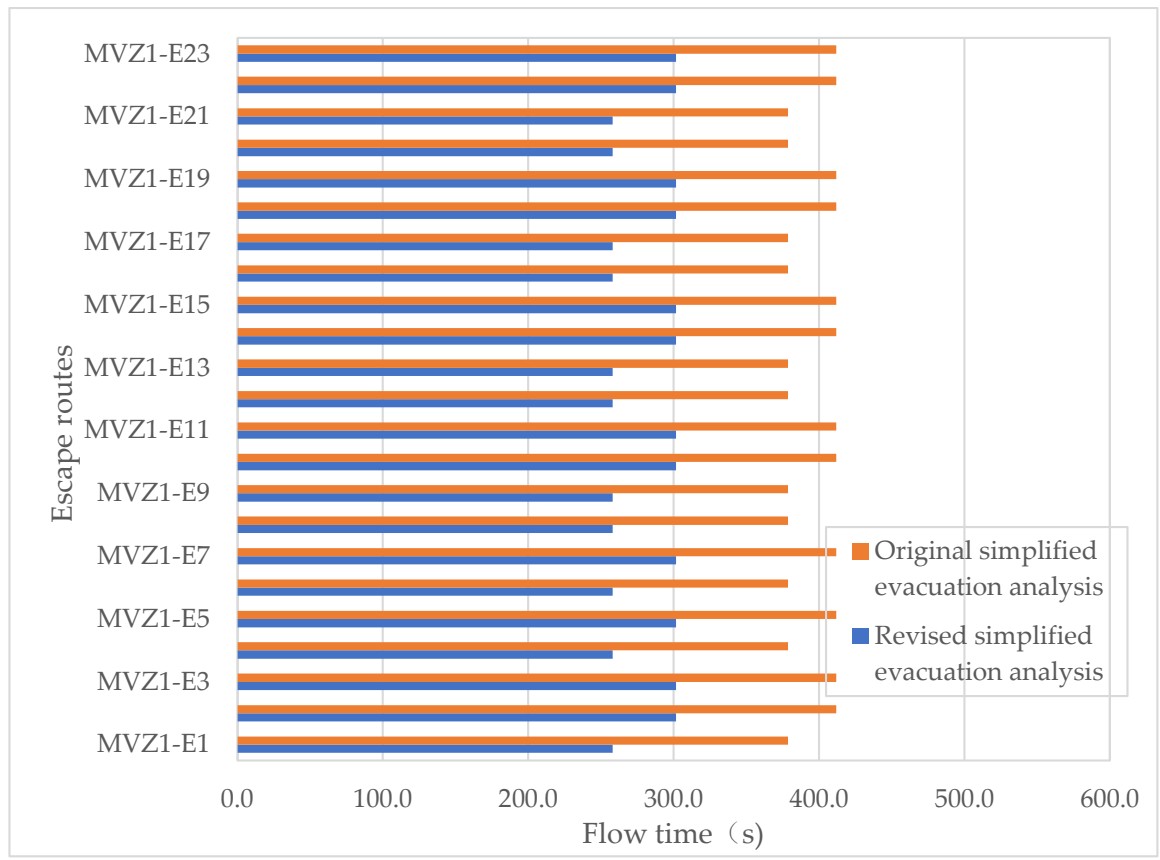

**Figure 6.** Comparison of flow time in MVZ 1/Original simplified evacuation analysis method: MSC/Cir.1533; Revised simplified evacuation analysis method: The proposed method in this paper.



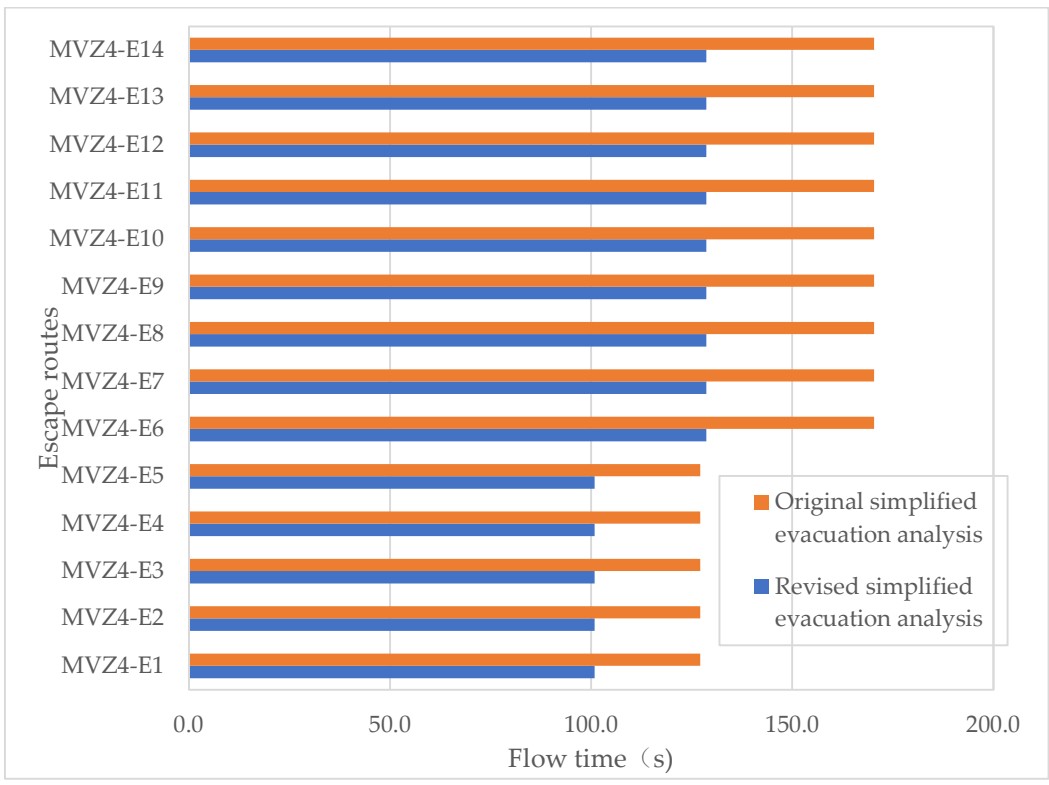

**Figure 7.** Comparison of flow time in MVZ 4/Original simplified evacuation analysis method: MSC/Cir.1533; Revised simplified evacuation analysis method: The proposed method in this paper.

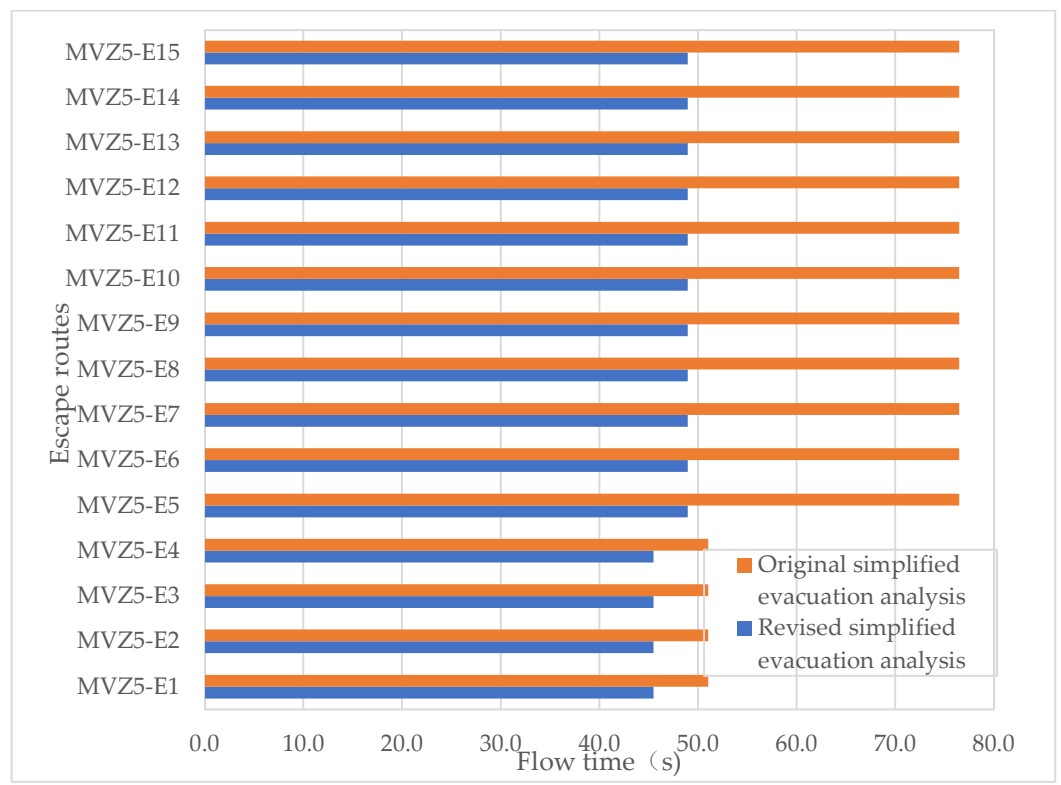

**Figure 8.** Comparison of flow time in MVZ 5/Original simplified evacuation analysis method: MSC/Cir.1533; Revised simplified evacuation analysis method: The proposed method in this paper.

### 4.3. Validations

To further verify the results of the proposed method, this paper takes the shipping platform as the object and uses the same ship layout scheme, passenger allocation scheme, assembly station location, and other environmental information as the input of the conventional method. Meanwhile, according to MSC.1/Circular.1533 [22], this simulation method of estimating the evacuation duration is based on the following assumptions: (i) The passengers and crew are represented as unique individuals with specified individual abilities and response durations; (ii) A safety factor having a value of 1.25 is introduced in the calculation to take account of model omissions, assumptions, and the limited number and nature of the benchmark scenarios considered; (iii) The population's composition is defined as shown in Tables 14 and 15; (iv) The response duration is not fixed for all the people, and the response duration distributions for the night-time scenario should be truncated logarithmic normal distributions as follows:

$$y = \frac{1.01875}{\sqrt{2\pi} \times 0.84 \times (x - 400)} \exp\left[-\frac{(\ln(x - 400) - 3.95)^2}{2 \times 0.84^2}\right] \tag{11}$$

**Table 14.** Population's composition of passengers.

| Population Groups—Passengers | Percentages (%) |
| --- | --- |
| Females < 30 years | 7 |
| Females 30–50 years old | 7 |
| Females > 50 years | 16 |
| Females > 50, mobility impaired (1) | 10 |
| Females > 50, mobility impaired (2) | 10 |
| Males < 30 years | 7 |
| Males 30–50 years old | 7 |
| Males > 50 years | 16 |
| Males > 50, mobility impaired (1) | 10 |
| Males > 50, mobility impaired (2) | 10 |

**Table 15.** Population's composition of crew.

| Population Groups—Crew | Percentages (%) |
| --- | --- |
| Crew females | 50 |
| Crew males | 50 |

This paper constructs the three-dimensional simulation model using the escape simulation analysis software maritime EXODUS. The simulation of MVZs 1, 4, and 5, are listed in Figure 9.

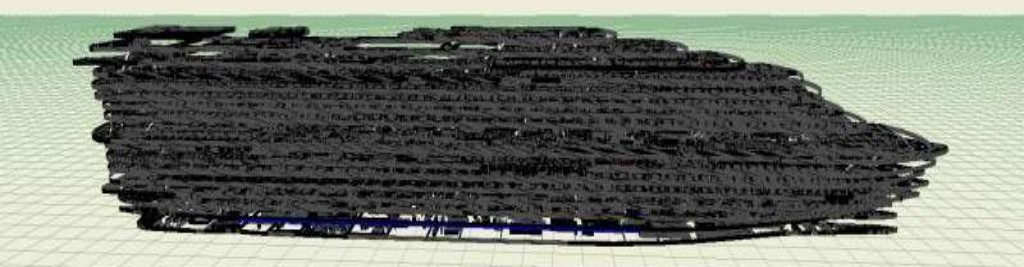

**Figure 9.** Simulation model of maritime EXODUS software.

Combined with the scenarios where passengers were located in various areas in the three main vertical areas, 50 simulations were carried out. Considering the uncertainties of the analysis, 95% of the simulated time is selected to represent all, which is marked with orange color in the Figures; see Figures 10–12.

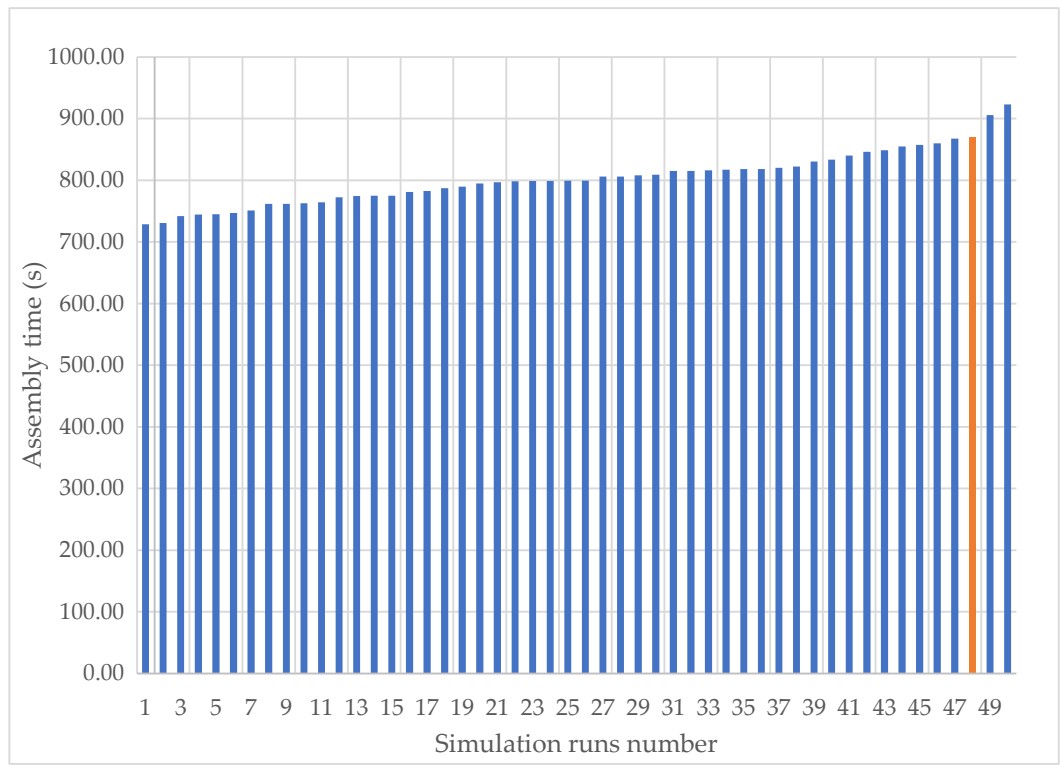

**Figure 10.** Convergence result of advanced evacuation analysis in MVZ 1.

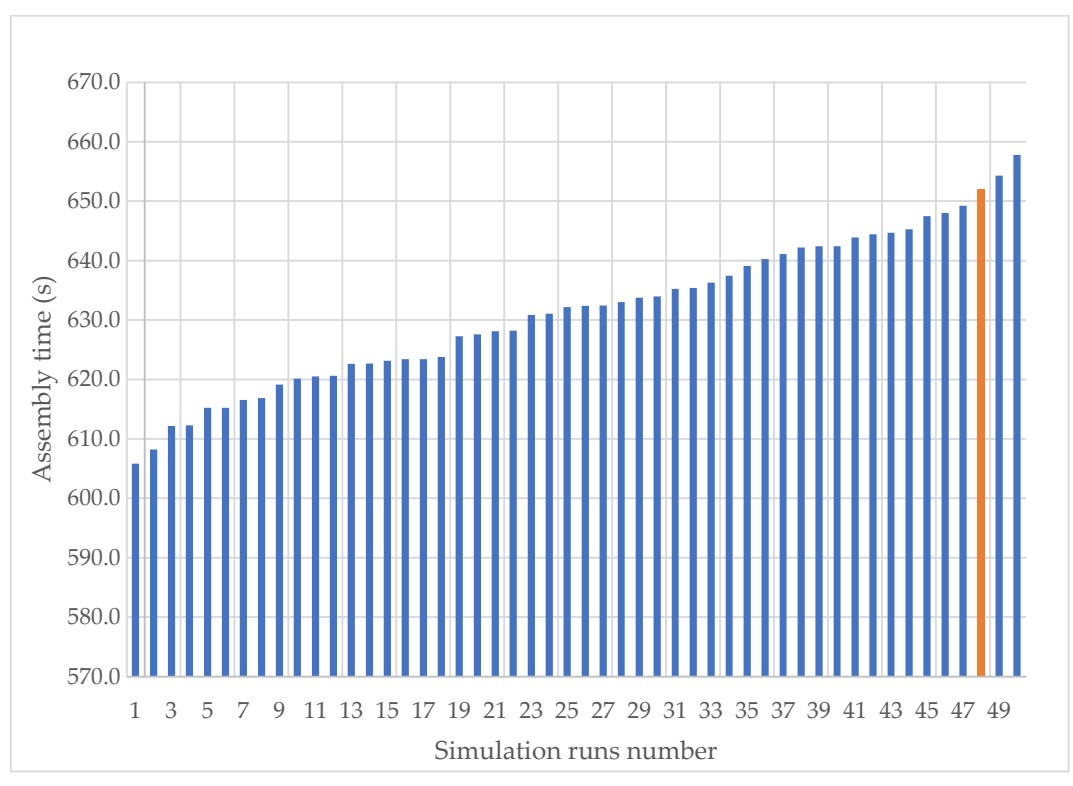

**Figure 11.** Convergence result of advanced evacuation analysis in MVZ 4.

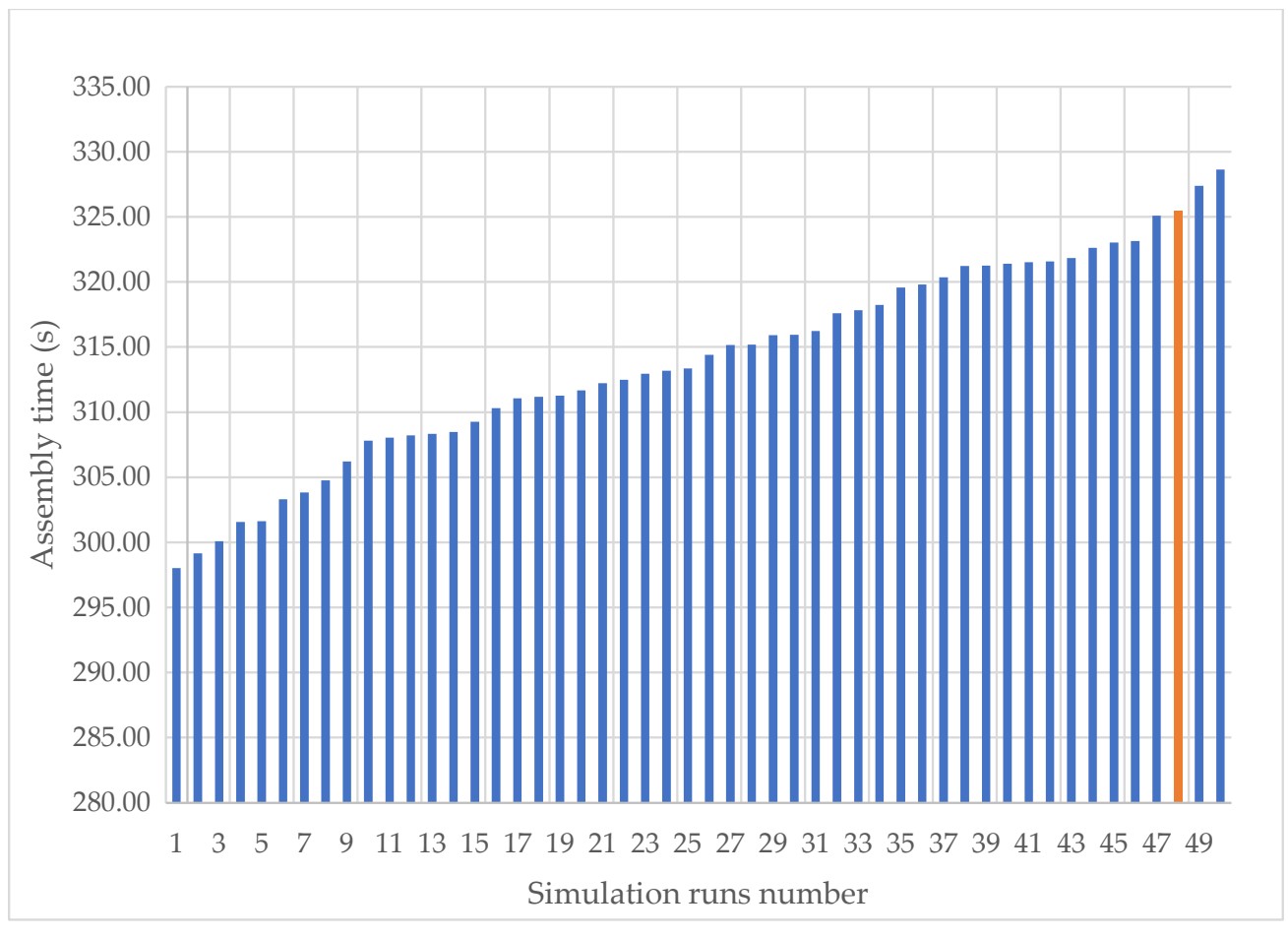

**Figure 12.** Convergence result of advanced evacuation analysis in MVZ 5.

The comparison results of travel durations between the MVZs computed by the proposed method, the traditional method, and the simulation tools are listed in Table 16. The comparison demonstrates that the deviations of the traditional model are 40.2% (MVZ 1), 9.7% (MVZ 4), and 21.9% (MVZ 5), respectively. The proposed method, however, reduced such deviations to 14% (MVZ 1), 0.2% (MVZ 4), and 11.9% (MVZ 5). It confirms the accuracy of the proposed method in this paper over the traditional one. Meanwhile, the time required for the preparation, calculation, and presentation of the results of the proposed method, the traditional method, and the simulation tools are listed in Table 17. The comparison demonstrates that the model-based method's efficiency is much higher than the simulation method, and the proposed method will only spend 4 h more on the evacuation analysis.

**Table 16.** Travel duration of MVZ 1/4/5 (s).

| MVZ | The Proposed Method (s) | Traditional Method (s) | Simulation Method (s) |
|---|---|---|---|
| MVZ 1 | 993.2 | 1221.0 | 871.2 |
| MVZ 4 | 650.9 | 715.1 | 652.1 |
| MVZ 5 | 364.2 | 396.8 | 325.5 |

**Table 17.** Time required for evacuation analysis.

| | The Proposed Method (Hours) | Traditional Method (Hours) | Simulation Method (Hours) |
|---|---|---|---|
| Time required for evacuation analysis | 64 | 60 | 144 |

## 5. Conclusions

This paper proposes an escape safety assessment method for cruise ships based on dislocation accumulation and social force models. First, personnel attributes, moving speed, and passenger flow are quantified by considering the age distribution of passengers; Subsequently, a new dislocation accumulation method is proposed to improve the confluence flow algorithm in stairways, by which the evacuation time and the movement speed of passengers in stairways are computed; Moreover, a flow time algorithm in congestion state is developed based on the social force model. A comparative analysis between the traditional escape evaluation method and the proposed one is conducted and is compared to the results of simulation tools to validate the results of the proposed method. The comparison results confirm that the proposed method holds the capability of computing more accurate results. Overall, the method proposed in this paper contributes to the safe designs of larger cruise ships.

The following points are for future works that follow the proposed evacuation time model: (i) With the assistance of more movement data collected from real ships under several emergency scenarios, a more accurate model can be constructed; (ii) The composition of passengers and crew members on passenger ships is dynamic, and more personnel attributes should be considered in the modeling; (iii) Consider people's countercurrent behavior into the model construction.

**Author Contributions:** Conceptualization, J.L. and G.W.; methodology, Y.G.; software, C.L.; validation, Y.H., J.L. and G.W.; resources, G.C.; writing—original draft preparation, J.L.; writing—review and editing, G.W.; visualization, Y.G.; supervision, G.C. All authors have read and agreed to the published version of the manuscript.

**Funding:** This research was funded by Shanghai Talent Development Fund, grant number 2019031.

**Institutional Review Board Statement:** Not applicable.

**Informed Consent Statement:** Not applicable.

**Data Availability Statement:** Not applicable.

**Conflicts of Interest:** The authors declare no conflict of interest.

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
