# Peer review of "Assessment of Escape Safety of Cruise Ships Based on Dislocation Accumulation and Social Force Models"

_applsci, doi:10.3390/app12167998_

Round 1

Reviewer 1 Report

The paper focuses on assessment of travel time during large ship evacuation. The authors introduce two elements: dislocation accumulation and social force model to fill the gap between model-based models and simulation-based approach. Literature review and methodological workflow are clear. Main weakness is lack of theoretical underpinning in the methodology – there is no review of the dislocation accumulation and social force models – are these new models or have been applied to other domains? Some expressions, like ‘more personnel distribution’ are not precise and unclear. Also, the differentiation between model-based methods and simulation-based methods is not convincing. Limitations and future research are missing.

Line 28-29: the root reason, please add references for this fundamental sentence. Moreover, the accident has not been caused by the escape/evacuation process that is posterior to the crash.

Line 30-32: this sentence is not clear and should be rewritten

Line 40 More recently

Line 41-42: too many repetitions of ‘simulation’

Line 43-44 this sentence is unclear, please rewrite

Line 55: ‘More developed…’ the subject here is unclear, maybe ‘more scholars’

Line 57-59: this sentence is not clear, please rewrite and explain the meaning of the term ‘solidification’ applied to the context of safety assessment methods

Line 70: please explain the meaning of convergence process

Line 216: please provide references for local rigidity theory

Line 217: please provide references for personnel flexibility theory

Line 223: why are the escape distances longer in nighttime scenarios?

Reviewer 2 Report

Assessment of escape safety of cruise ships based on dislocation accumulation and social force models

Please find my remarks below:

Line 10 – please check grammar

Lines 45 and 46 – please check grammar

Line 26 – I’m not sure if word “always” is correct. Not all accidents are catastrophic.

Lines 62-84 – very interesting summary of existing models

Line 123 – described assumption is not realistic – but can have little influence to the final results due to the congestion. Please consider some commenting on that – to defend this assumptions.

Lines 130-140 – Typically the flow is described as a number of people crossing a line in a specific time. This line should have a width. In the text there is an information about area. Please check the units. Shouldn’t it be something like [ppl/(m*s)] ?

Line 141 – I’m assuming that in this case you are calculating time = not flow?

Lines 169 – 171 – Please add sources

Line 185 – The units are mixed. The table is not clear. Does the specific flow refer to speeds or vice versa?

Line 192 – What is the unit for Fs – if it is as it is described in table 8 – then – the unit of tF-stair won’t be (s)…

Line 225 – typically this coefficient (1.3) has a unit.

Overall there is some confusion in units and naming of time and flow. Please consider a double check in this area.

Line 233 – please consider adding some figures of the ship – for a reader it would be very helpful

Line 243 – If the night time scenario was selected – please consider some discussion on the response time.

Line 251 – Please consider using brackets for units in header of the table 11.

Line 251 – The “Item” column is not clear – what does it mean ?

Line 252 – Please add units

Line 286 – The legend for the figure 5 is illegible.

Line 297 – Comment on the figures 5, 6, 7. Please consider the X axis staring from 0. The should be some quantified summary on the differences between models.

Line 298 – The is not enough date on input data to the simulation. What were the initial assumptions, the response times, populations groups and characteristics?

Line 323 – Please consider adding a comment od the time required for preparation, calculation and presentation of the results of both – model-based and simulation-based models.

Line 328 – typo – “tine”

Overall conclusion – proposed calculation model is very interesting. But it’s validation should be reconsidered. The validation is conducted only on one ship with very high number of people. In this case it is possible that only one factor is enough to asses the assembly time – the final bottleneck and the distance from it to the muster station. Is there a possibility to present a sensitivity analysis – which areas or process is responsible for the time duration?

The response times were omitted for a night scenario. Unless it is a standard from IMO – there should be more discussion provided on that matter.

And finally the title – this paper is about assessment of the evacuation time not about an assessment of safety. Please reconsider change in the title.
